# Broad and direct interaction between TLR and Siglec families of pattern recognition receptors and its regulation by Neu1

Guo-Yun Chen[1]*, Nicholas K Brown[1], Wei Wu[1], Zahra Khedri[2], Hai Yu[2], Xi Chen[2], Diantha van de Vlekkert[3], Alessandra D'Azzo[3], Pan Zheng[1,4]*, Yang Liu[1]*

[1]Center for Cancer and Immunology Research, Children's National Medical Center, Washington, DC, United States; [2]Department of Chemistry, University of California, Davis, Davis, United States; [3]Department of Genetics, St Jude Children's Research Hospital, Memphis, United States; [4]Division of Pathology, Children's National Medical Center, Washington, DC, United States

**Abstract** Both pathogen- and tissue damage-associated molecular patterns induce inflammation through toll-like receptors (TLRs), while sialic acid-binding immunoglobulin superfamily lectin receptors (Siglecs) provide negative regulation. Here we report extensive and direct interactions between these pattern recognition receptors. The promiscuous TLR binders were human SIGLEC-5/9 and mouse Siglec-3/E/F. Mouse Siglec-G did not show appreciable binding to any TLRs tested. Correspondingly, *Siglece* deletion enhanced dendritic cell responses to all microbial TLR ligands tested, while *Siglecg* deletion did not affect the responses to these ligands. TLR4 activation triggers Neu1 translocation to cell surface to disrupt TLR4:Siglec-E interaction. Conversely, sialidase inhibitor Neu5Gc2en prevented TLR4 ligand-induced disruption of TLR4:Siglec E/F interactions. Absence of *Neu1* in hematopoietic cells or systematic treatment with sialidase inhibitor Neu5Gc2en protected mice against endotoxemia. Our data raised an intriguing possibility of a broad repression of TLR function by Siglecs and a sialidase-mediated de-repression that allows positive feedback of TLR activation during infection.

**\*For correspondence:** gchen@cnmc.org (G-YC); pzheng@cnmc.org (PZ); yaliu@cnmc.org (YL)

**Competing interests:** The authors declare that no competing interests exist.

**Reviewing editor**: Xuetao Cao, Zhejiang University School of Medicine, China

## Introduction

Nearly 25 years ago, Janeway proposed that the innate immune system discriminates infectious nonself from non-infectious self through non-clonally distributed pattern recognition receptors (PRRs) (*Janeway, 1989, 1992*). This concept was supported by the observations of microbial induction of costimulatory activity on antigen-presenting cells (*Liu and Janeway, 1991, 1992*; *Wu and Liu, 1994*) and bolstered as a major pillar in immunology by the identification of TLRs (*Medzhitov et al., 1997*) and its ligands (*Poltorak et al., 1998*). Over the following decades, TLRs have emerged as a family of PRRs that sense a variety of pathogen-associated molecular patterns (PAMPs), ranging from microbial glycan, bacterial glycolipids, flagellin, and viral and bacterial nucleic acids (*Kawai and Akira, 2010*).

Surprisingly, accumulating data demonstrated that TLRs also sense cellular components released after cellular injuries, such as heat-shock proteins (*Millar et al., 2003*), HMGB1 (*Apetoh et al., 2007*; *Ivanov et al., 2007*; *Tian et al., 2007*) and S100 (*Hiratsuka et al., 2008*; *Tsai et al., 2014*). These components were collectively called DAMPs for danger-associated molecular patterns, based on Matzinger's 'danger theory' (*Matzinger, 1994*). In addition to TLRs, Nod-like receptors (NLRs) have also been shown to respond to both microbial components (*Franchi et al., 2012*) and cellular injuries (*Ting et al., 2008*; *Tschopp and Schroder, 2010*). Since these PRRs respond to both infectious and noninfectious inflammatory stimuli, additional regulatory mechanism was deemed

**eLife digest** Many living things have an immune system that is able to detect invading bacteria, viruses and other pathogens and trigger a response targeted against the threat before it causes lasting damage. Cells employ a number of different receptors that can detect these pathogens or the molecules that they produce.

In animals, toll-like receptors (or TLRs) are a type of protein that recognizes patterns or structures that are found in many different types of pathogen, known as pathogen-associated molecular patterns (or PAMPs). Injured cells release proteins that are also recognized by toll-like receptors and are called danger associated molecular patterns (or DAMPs). An immune response is triggered when PAMPs and DAMPs are recognized, but the response must be properly controlled. If it goes awry, it can result in an over-activation of the immune cells that can lead to life-threatening conditions, one of which is called sepsis.

Siglecs are proteins that bind to a sugar molecule, which is found attached to many other proteins, and are known to inhibit the immune response. However, it remained unclear how Siglecs do this and if they can interact directly with toll-like receptors. Chen et al. now show that most (although not all) Siglecs bind to TLRs, and that deleting the gene for a Siglec protein that can bind to multiple TLRs boosted the response of the immune cells to a range of microbial PAMPs. Deleting the gene for another Siglec that did not bind to any TLRs had no effect on the immune response.

Chen et al. suggest that the Siglec proteins that interact with toll-like receptors act a bit like a brake that slows down the activation of the receptors. However, when an immune cell detects a foreign molecule through a TLR, an enzyme called Neu1 is relocated from the inside of the cell to the cell's surface, where it removes the sugar molecules from the TLRs. This disrupts the interaction between the TLRs and the Siglecs, thus activating the receptors and triggering an immune response against the invading pathogen or damaged cells. This represents a newly discovered mechanism that can regulate the signaling of TLRs.

Chen et al. also show that a chemical compound that stops the function of the Neu1 enzyme prevents the toll-like receptors—and hence the immune cells—from becoming overly activated. Mice treated with this compound are protected against sepsis triggered by the presence of a bacterial PAMP. These results suggest that the Neu1 enzyme may be a promising new target for treating sepsis; further work will now be required to assess the potential side effects caused by inhibiting this enzyme.

needed if pattern recognition receptors were to be the key components to discriminate the two (*Liu et al., 2009*).

We have reported that CD24-Siglec-G/10 interactions selectively inhibit host responses to DAMPs without affecting responses to PAMPs (*Chen et al., 2009*). These data suggest a new mechanism for limiting inflammation to autologous molecules, and a framework for integrating the self-nonself and danger theories of immunity (*Liu et al., 2009*).

Siglecs are membrane-bound lectins that constitute the sialic acid-binding immunoglobulin super family with distinct cellular distribution and glycan specificities (*Crocker et al., 2007*). Most Siglecs have intracellular domains capable of inhibitory signaling. With a few notable exceptions (*Stamenkovic et al., 1991*; *Chen et al., 2009*; *Bandala-Sanchez et al., 2013*; *McMillan et al., 2013*), the natural ligands for most Siglecs remain to be determined. The selective effect of Siglec-G in regulating host response to DAMPs raised two interesting questions: First, do all Siglecs perform similar functions in discriminating self from non-self in innate immunity? Second, do Siglec family members directly interact with other families of pattern recognition receptors and play a broad function in regulating innate immunity? Here we addressed these two issues by revealing a broad and direct interaction between TLRs and many Siglecs, and by highlighting Siglec-G for its lack of direct interactions with TLR.

A standard test to validate Siglec-mediated interactions is to measure their susceptibility to sialidase treatment (*Crocker et al., 2007*). The implication of this cardinal feature in immune recognition has not been widely explored. Our previous studies have demonstrated that CD24-Siglec-G interactions are disrupted by bacterial sialidase and that such disruptions exacerbates sepsis (*Chen et al., 2011*). Since most pathogens do not express sialidases, the implication of this observation on immune

regulation during infection and autoimmune diseases is less clear. On the other hand, mammals express at least four sialidases, Neu1-4 (*Miyagi and Yamaguchi, 2012*). Several studies have suggested that activation of TLR can be attenuated by sialidase inhibitors (*Amith et al., 2010*; *Abdulkhalek et al., 2011*). Both Neu1 and Neu3 have been suggested as targets of the inhibitors (*Amith et al., 2010*; *Abdulkhalek et al., 2011*). Neu1 has been shown to cause TLR4 dimerization, which was suggested to be important for its regulation of TLR4 activation (*Amith et al., 2010*). Here we report a critical role for Neu1 in regulating Siglec-TLR interaction and endotoxemia. Together, our data reveal an overlooked network of interactions among Siglecs, host sialidases and TLRs and, with more potent Neu1 inhibitors, propose host sialidases as therapeutic targets for lethal endotoxemia.

## Results

### Extensive Siglec-TLR interactions negative regulate the function of multiple TLRs

To test whether TLRs directly interact with Siglecs, we first tested a panel of recombinant human SIGLEC fusion proteins for their interaction with human TLRs synthesized by the THP1 cell line using a sandwich capture assay. Plates were coated with recombinant SIGLEC-Fc fusion proteins or control human Fc. Lysates from the human myeloid cell line THP1 was used as the source of cellular TLRs, as this line expresses transcripts of all TLRs at significant levels (*Figure 1A*). SIGLEC-Fc-bound TLRs were detected with anti-TLR antibodies. This assay revealed extensive interaction between the two families of pattern recognition receptors (*Figure 1B*). Among them, Siglec 5 and 9 cross-reacted with virtually all TLR tested. In contrast, very little interaction was observed between TLRs and SIGLECS-1, 2, 7, 10 and 11.

Since the anti-TLR antibodies used also cross-react with their mouse homologs, we carried out a similar analysis between mouse Siglecs and Tlrs. As shown in *Figure 1C*, Siglec-3, E, F and H were broadly cross-reactive, while Siglec-G displayed no interaction with any murine Tlr. We validated the interaction between Siglec-E and Tlr4 using both pull down and bi-directional immunoprecipitation assays. Siglec-E-Fc, but not control IgG, pulled down Tlr4 in the spleen cell lysates (*Figure 1D*). Anti-Siglec-E monoclonal antibodies co-precipitated Tlr4, while anti-Tlr4 co-precipitated Siglec-E (*Figure 1E*). To determine whether Siglecs directly interact with TLR, we coated the 96-well plates with recombinant Siglecs or control IgG. After blocking with bovine serum albumin, recombinant TLR4 extracellular domain was added, and the bound TLR4 were determined using biotinylated anti-TLR4 and horseradish peroxidase-conjugated streptavidin. The data revealed that TLR4 directly interact strongly with SIGLEC-5, 6, 9, 11 and modestly with SIGLEC-7 (*Figure 1F*). In contrast, no interaction was observed between SIGLEC-1, 2, 3, 10 and TLR4. Likewise, mouse Siglec-E and F bound strongly to TLR4, with Siglec-3 and H showing modest binding to TLR4, and no binding to TLR4 and Siglec-1, 2 and G (*Figure 1G*). Comparisons between *Figure 1A,E*, and between *Figure 1C,G*, show that, with the modest exception of Siglec-7, recombinant TLR4 recapitulated the specificity of endogenous TLR4. Taken together, data in *Figure 1* demonstrate broad and direct interaction between TLR and Siglec families of PRRs.

Since Siglec-E contains intracellular immunoreceptor tyrosine inhibitory motif (ITIM) and ITIM-like domains and associates with molecules known to negatively regulate production of inflammatory cytokines, we compared bone marrow-derived dendritic cells (DC) from WT and *Siglece*−/− mice for their responses to prototypic TLR ligands. After overnight stimulation with TLR ligands, we analyzed the production of IL-6 and TNFα by DC. As shown in *Figure 2A*, *Siglece*−/− DC were 1000-fold more responsive to LPS stimulation than *Siglece*+/+ DC. Likewise, *Siglece*−/− DC produced at least 10-fold more cytokines in response to CpG (*Figure 2B*). Given the multiple interactions of Siglec-E with other TLR (*Figure 1C*), we tested if endogenous Siglec-E negatively regulates production of inflammatory cytokines to other TLR ligands. As shown in *Figure 2C*, in addition to an enhanced response to TLR4 and TLR9 ligands, *Siglece*−/− DC also produced significantly more IL-6 in response to synthetic triacylated lipoprotein Pam3CSK4 (TLR1/2 agonist), heat-killed *Listeria monocytogenes* (HKLM, Tlr2 agonist), poly(I:C), (TLR3 agonist), *Salmonella typhimurium* flagellin (ST-FLA, TLR5 agonist), synthetic lipoprotein derived from *Mycoplasma salivarium* (FSL-1, TLR2/6 agonist) and ssRNA40, a 20-mer phosphorothioate-protected single-stranded RNA oligonucleotide containing a GU-rich sequence (TLR8 agonist). Since Siglec-E negatively regulates responses to all TLR ligands tested, we suggest that the physical interactions between Siglec-E and TLRs are biologically significant. Except for a modest elevated

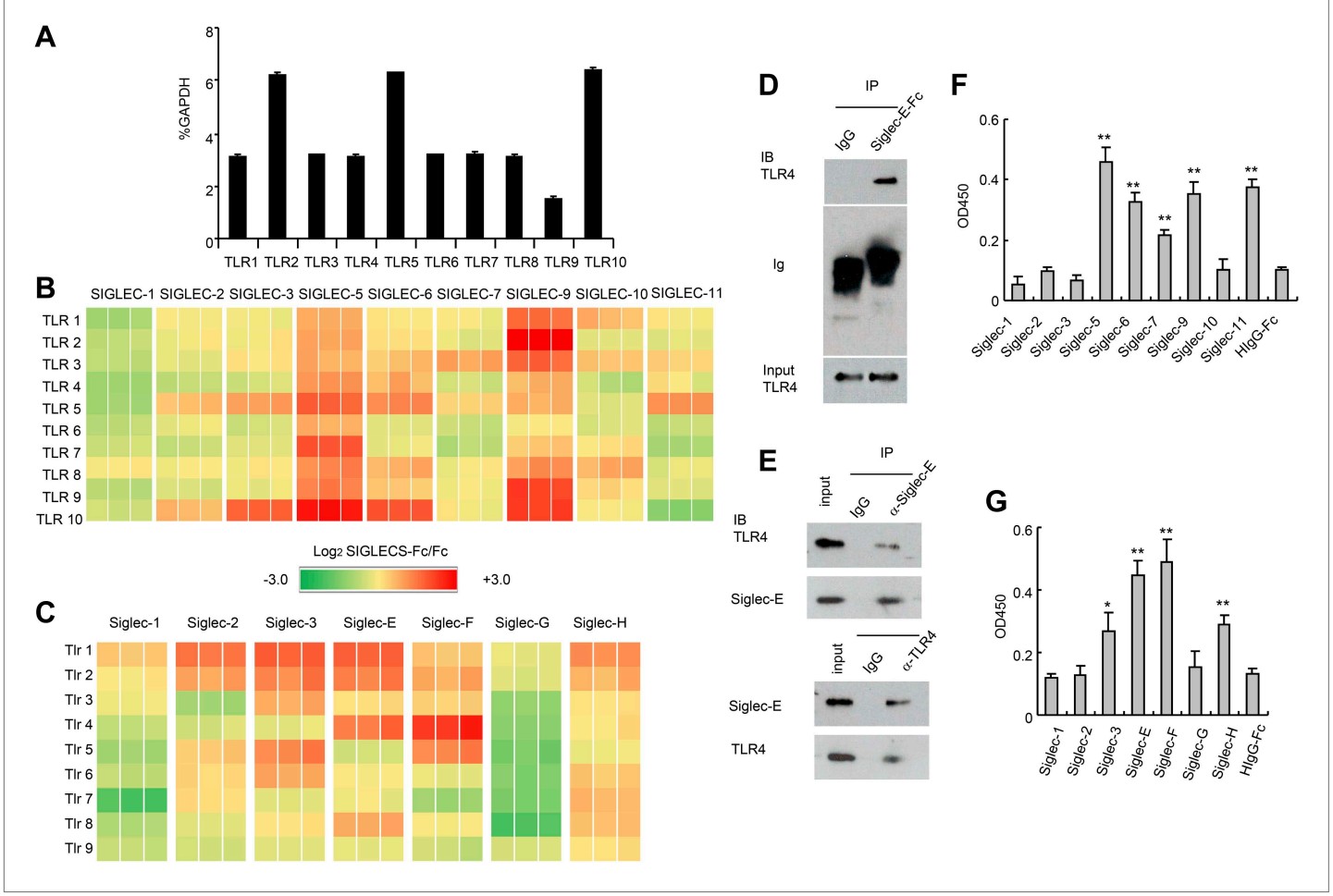

**Figure 1**. Extensive direct interactions between Siglecs and TLRs. (**A**) Evaluation of TLR expression using TLR-primer set. Data shown are means and SEM of triplicate % of *GAPDH* levels. (**B**) and (**C**) Interactions between human (**B**) or mouse (**C**) SIGLEC-Fc fusion proteins and TLRs from THP-1 cells (**B**) or murine splenocyte lysates (**C**). Recombinant SIGLEC-Fc or control IgG Fc were coated on 96 well plates to capture TLR in the cell lysates. The associated TLRs were detected with biotinylated anti-TLR antibodies that cross-react with both mouse and human TLR. Data shown are the log2 ratios between Siglec-Fc and IgG Fc in triplicate and were repeated three times. (**D**) Recombinant Siglec-E binds to endogenous Tlr4. Lysates from C57BL/6 mouse splenocytes were incubated with either Fc control or SIglec E-Fc. After precipitation with protein A beads, the precipitates were analyzed by Western blot, using antibodies against Siglec-E, Tlr4, and Fc. (**E**) Interaction between endogenous Tlr4 and Siglec-E in D2SC dendritic cells. Lysates from D2SC cells were immunoprecipitated with either anti-Siglec-E (top) or Tlr4 (bottom) antibodies. The precipitates were analyzed by Western blot, using antibodies against Siglec-E or Tlr4. Similar results were obtained when performed with WT mouse splenocyte lysates (data not shown). (**F**) Direct interaction between human Siglecs and ectodomain of TLR4. As in (**A**), except the cell lysates were replaced with recombinant TLR4. (**G**) Direct interaction between mouse Siglecs and ectodomain of TLR4. As in (**B**), except the cell lysates were replaced with recombinant TLR4. Data presented in this figure have been reproduced at least three times.

response to ssRNA40, *Siglecg*$^{-/-}$ and *Siglecg*$^{+/+}$ DC exhibited similar responses to all TLR ligands tested (*Figure 2C*). This, along with data from our previous report indicating that Siglec-G does not inhibit inflammatory response to LPS and poly(I:C), (*Chen et al., 2009*), is consistent with the lack of physical interactions between Siglec-G and TLRs (*Figure 1C,G*).

## The TLR4-Siglec-E interaction is negatively regulated by Neu1

We next tested the impact of LPS stimulation on Siglec-E-TLR4 interaction. As shown in *Figure 3A*, co-precipitation between endogenous TLR4 and Siglec-E was substantially reduced after LPS stimulation. These data demonstrate a dynamic regulation of TLR4-Siglec-E association on DC. Since interaction of Siglecs with their ligands is dependent on sialic acid and can be disrupted by sialidase (*Crocker et al., 2007*), and since LPS is devoid of sialidase activity, we evaluated the contribution

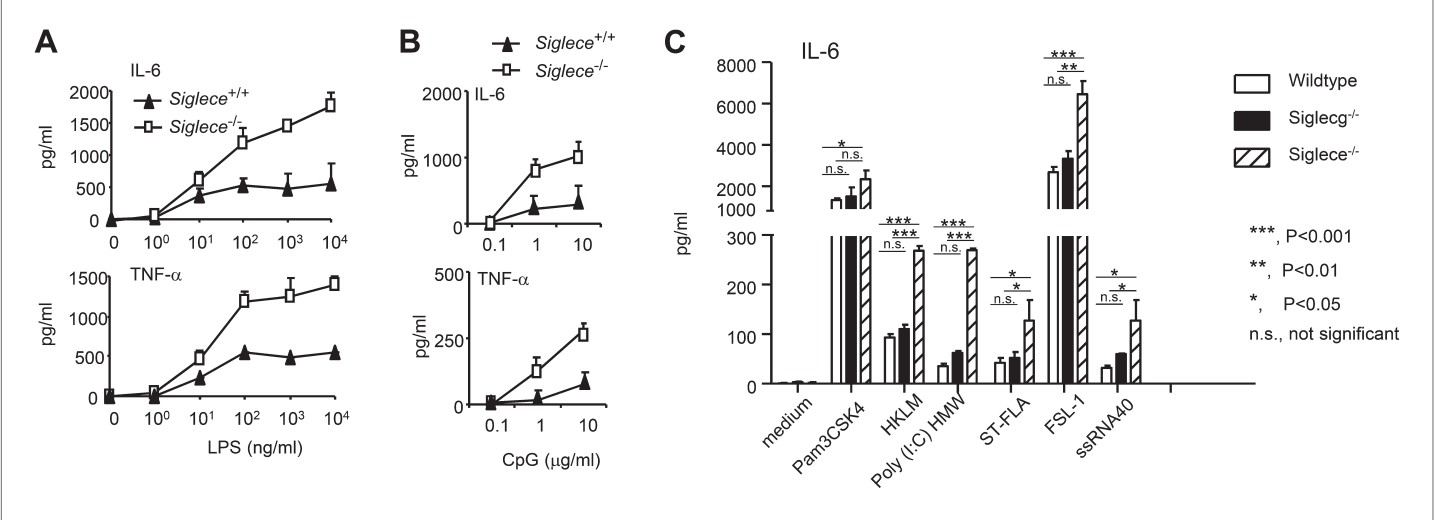

**Figure 2**. Siglec-E negatively regulates production of inflammatory cytokines by DC in response to TLR ligands. (**A**) and (**B**) Siglec-E inhibits production of IL-6 and TNFα by bone marrow derived DC. DC cultured from WT or *Siglece⁻/⁻* bone marrow were stimulated with indicated concentrations of LPS (**A**), or poly(I:C) (**B**) for 16 hr, and supernatant cytokine concentrations were analyzed with cytokine bead array. (**C**). Targeted mutation of Siglec-E, but not Siglec-G, enhances production of IL-6 to multiple TLR ligands. The TLR agonists used are: synthetic triacylated lipoprotein Pam3CSK4 (TLR1/2 agonist), heat-killed *Listeria monocytogenes* (HKLM, Tlr2 agonist), poly(I:C), (TLR3 agonist), *Salmonella typhimurium* flagellin (ST-FLA, TLR5 agonist), synthetic lipoprotein derived from Mycoplasma salivarium (FSL-1, TLR2/6 agonist) and ssRNA40, a 20-mer phosphorothioate-protected single-stranded RNA oligonucleotide containing a GU-rich sequence (TLR8 agonist). All agonist were used at 100 ng/ml. Data represent the mean ± SD for three independent cultures of DCs in each genotype and were repeated at least three times.

of endogenous sialidase, using the D2SC DC cell line (*Bachmann et al., 1996*). Real-time PCR analysis indicated that D2SC cells express *Neu1* and to a lesser extent *Neu3*, but not *Neu2* or *Neu4* (*Figure 3B*). Since TLR4 is a cell surface glycoprotein while Neu1 is primarily lysosomal, we evaluated whether Neu1 translocates to the cell surface following LPS stimulation. Fluorescent microscopy revealed a robust translocation of Neu1 to cell surface where it co-localized with TLR4 (*Figure 3C*), which is similar to a previous report on macrophages (*Liang et al., 2006*). Flow cytometry confirmed a time-dependent translocation of Neu1 between 6–18 hr following LPS stimulation (*Figure 3D*). To test if Neu1 and TLR4 interact with each other, live cells were cross-linked with dithiobis[succinimidyl propionate ] (DSP) to stabilize transient enzyme–substrate interactions. After cross-linking, the LPS-treated and untreated D2SC cells were lysed for co-immunoprecipitation. As shown in *Figure 3E*, a specific Neu1-TLR4 association was observed only after LPS stimulation.

To determine if Neu1 regulates cell surface Siglec-E ligand levels, we compared binding of Siglec-E-Fc to either scrambled or *Neu1* shRNA-transduced D2SC cells. As shown in *Figure 3F*, silencing of *Neu1* increased binding of Siglec-E-Fc to DC. To determine whether Neu1 contributes to LPS-induced disassociation between Siglec-E and TLR4, we tested the impact of *Neu1* silencing on Siglec-E-TLR4 interaction, as measured by co-immunoprecipitation. As a control, we also tested the impact of silencing *Neu3*, which is a constitutive cell surface sialidase. As shown in *Figure 3G*, shRNA silencing of *Neu1* abrogated the LPS-induced disassociation of Siglec-E-TLR4 complexes. In contrast, shRNA silencing of *Neu3* had no effect on Siglec-E-TLR4 interaction.

To determine whether Neu1 regulates production of inflammatory cytokines, we tested three independent *Neu1* shRNAs (*Neu1sh1-3*)-silenced D2SC cell lines for their responses to LPS. As shown in *Figure 3H*, the three shRNAs reduced cell surface Neu1 in LPS-stimulated D2SC cells with different efficiencies: Neu1 was completely silenced by Sh1 and Sh2, while only partially suppressed by Sh3. Corresponding with the silencing efficiencies, LPS-induced cytokine production was more significantly reduced by Sh1 and Sh2 than by Sh3, which had partial effect when compared to scramble control (*Figure 3I*). Taken together, the data presented in this section demonstrate dynamic regulation of the TLR-Siglec-E interaction by Neu1 and its impact on DC response to LPS, the prototypic TLR4 ligand.

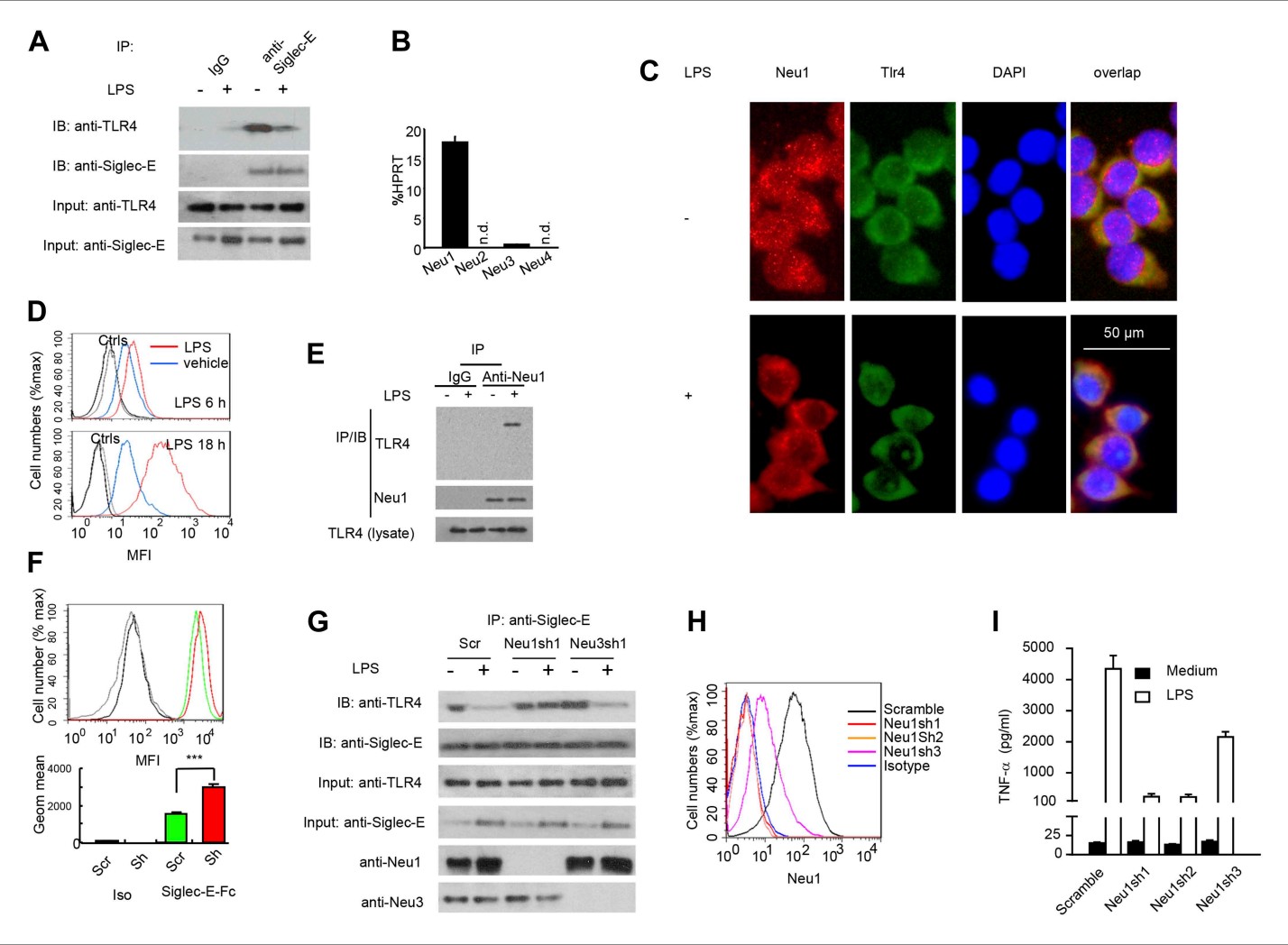

**Figure 3**. A critical role for Neu1 in Tlr4 activation. (**A**) Siglec-E-Tlr4 association is disrupted by LPS stimulation. D2SC cells were cultured in the presence or absence of LPS overnight. Immunoprecipitation was used to test Siglec-E-Tlr4 association as detailed in the *Figure 1D* legend. (**B**) Expression of *Neu1-4* mRNA in D2SC dendritic cells was determined by RT-PCR. Data shown are mean ± SD transcript levels, expressed as % of the housekeeping gene HPRT. (**C**) Translocation of Neu1 and its co-localization with TLR4 in D2SC cells. D2SC were cultured in the presence or absence of LPS (100 ng/ml) for 18 hr and co-stained with anti-TLR4 and anti-Neu1 antibodies. (**D**) Increased cell surface expression of Neu1 on D2SC cells after stimulation with LPS as revealed by flow cytometry. D2SC cells were treated with LPS (100 ng/ml) or vehicle for 6 hr (upper panel) or 18 hr (lower panel). The expression of cell-surface Neu1 was determined by FACS. (**E**) Physical association between Neu1 and TLR4. D2SC 2 cell lines were stimulated with 2 μg/ml LPS or vehicle for 16 hr were crossed linked with 1 mM DSP at room temperature for 30 min. The lysates were immunoprecipitated with anti-Neu1 and then probed with anti-Neu1 or anti-TLR4. (**F**) Silencing Neu1 by lentivirus shRNA increased the cell surface Siglec-E ligands. Histograms shown on top panels are FACS profiles. The bar graphs in the bottom panels represent geometric means ± SD of fluorescence intensity (*n* = 3). (**G**) Neu1 disrupts Tlr4-Siglec-E association in DC. D2SC dendritic cells were transduced with lentiviral vector carrying scrambled shRNA, three independent Neu1 shRNAs or Neu3 shRNA. After LPS stimulation for 24 hr, the lysates were used for immunoprecipitation. (**H**) ShRNA silencing of Neu1 affect cell surface Neu1 levels. Data shown are histogram of flow cytometry data depicting cell surface expression of Neu1 in LPS-stimulated D2SC clones. (**I**) An essential role for Neu1 in production of TNFα by D2SC cells. Aliquots of 2 × 10^5 D2SC transfectants were stimulated with LPS (100 ng/ml) for 12 hr. The culture supernatants were subsequently collected and analyzed for TNF-α. Scramble, Neu1sh1, 2, 3 represent stable clones expressing three independent Neu1 ShRNAs. Experiments depicted in this figure have been reproduced two to three times.

## *Neu1* deletion in hematopoietic cells confers resistance to endotoxemia

To confirm the impact of LPS stimulation on Neu1 translocation and sialylation of primary leukocytes, we injected intraperitoneally (i.p.) sublethal doses of LPS (200 μg/mouse) or PBS control into C57BL/6 mice, and analyzed spleen cells 16 hr later for cell surface expression of Neu1. As shown in *Figure 4A*, LPS stimulation led to upregulation of cell surface Neu1 on DCs, macrophages, and neutrophils.

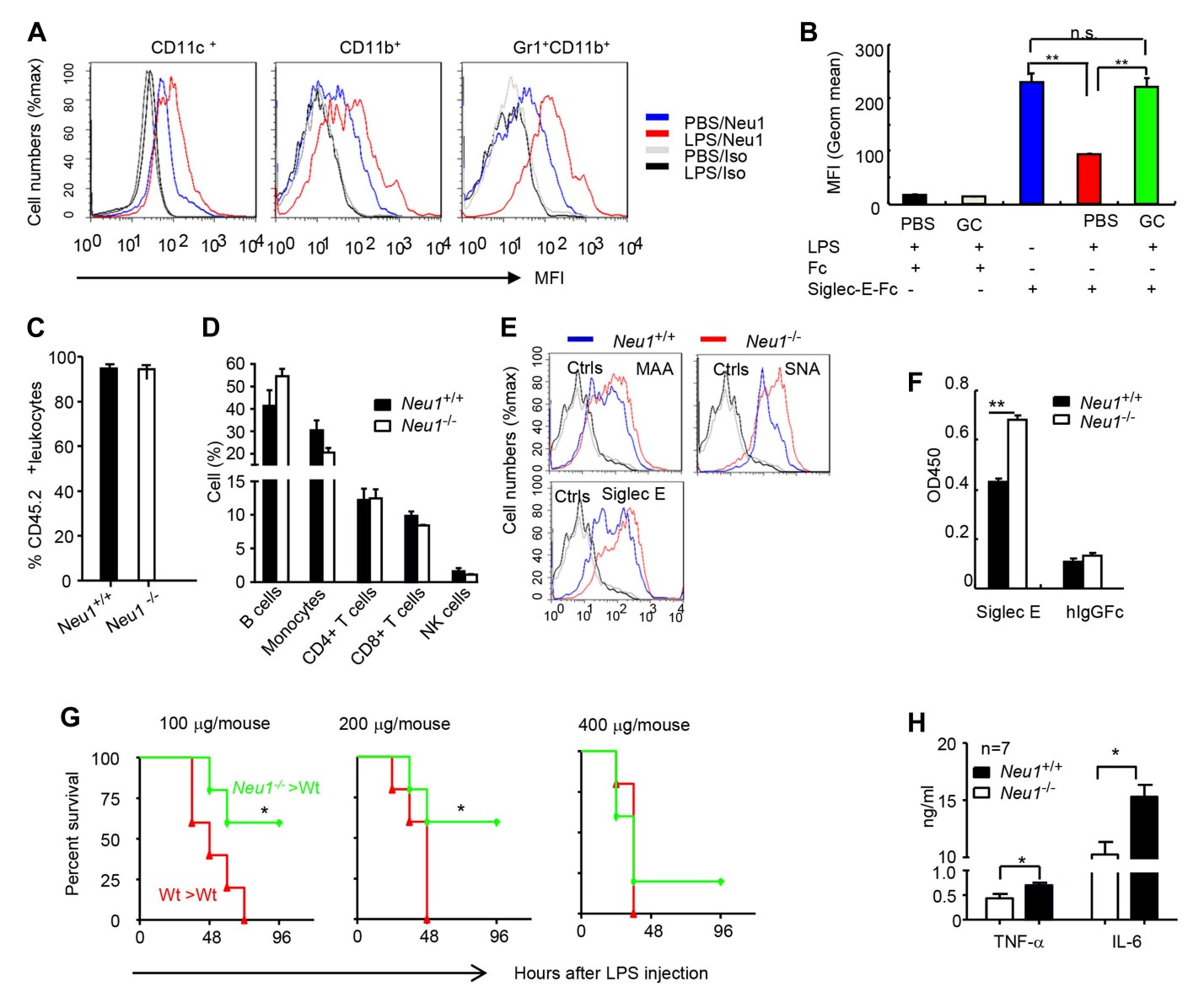

**Figure 4**. A critical role for hematopoietic cell-expressed Neu1 in endotoxic shock. (**A**) LPS stimulation in vivo increased cell surface Neu1 on DC, macrophage, and neutrophil. Data are representative of those from two independent experiments involving two mice per group. Splenocytes were collected from mice 16 hr after they received an injection of LPS (i. p. 200 μg/mouse). (**B**) The cell surface Siglec-E ligands are down-regulated on DC following LPS stimulation. Data shown are means ± SD (n = 3) and have been reproduced twice. (**C–H**) Lethally irradiated CD45.1 congenic B6 mice were transplanted with WT or *Neu1⁻/⁻* BM (5 × 10⁶ cells/mouse). 21 weeks later, the mice were bled to analyze hematopoiesis. (**C**) Comparable reconstitution of donor-derived hematopoietic cells, based on the frequencies of donor-derived CD45.2⁺ cells. (n = 5). (**D**) Normal CD45.2⁺ leukocyte composition of PBL as determined by flow cytometry. Populations were defined as: B cells, B220⁺; monocytes, NK1.1⁻CD11b⁺; CD4⁺ T cells, CD4⁺CD3⁺; CD8⁺ T cells, CD3⁺CD8⁺; NK cells, NK1.1⁺. (**E**) Increase of α2,3 and α2,6 sialylation and Siglec-E ligand on *Neu1⁻/⁻* DC. (Gated CD11c⁺ as DCs). Splenocytes from *Neu1⁺/⁺* and *Neu1⁻/⁻* mice were stained with FITC-conjugated SNA, MAA, or unconjugated Siglec-E-Fc (detected with PE-anti-mouse IgG Fc) in conjunction with APC-conjugated anti-CD11c mAb. Data shown are histograms depicting the binding of SNA, MAA and Siglec-E-Fc respectively. (**F**) Siglec-E-Tlr4 association is increased on *Neu1⁻/⁻* splenocytes, as measured by captured Tlr4 in plates coated with Siglec-E-Fc. (**G**) *Neu1* deficiency increases resistance to LPS challenge. Data shown are Kaplan Meier survival curves of mice that received the indicated doses of LPS (i.p., n = 5 in all groups). The mice were observed for 2 weeks, though all death occurred within 72 hr. (**H**) Cytokine production in blood measured 16 hr after LPS treatment. Data represent the mean ± SD. All data in this figure are representative of two to three independent experiments.

Correspondingly, Siglec-E-Fc showed reduced binding to DCs after LPS stimulation, and this effect was rescued by incubation with a sialidase inhibitor, indicating that Siglec-E binding to its ligands was disrupted by sialidase activity (*Figure 4B*).

Since we showed Neu1 to be a key regulator of Siglec-E-TLR4 interaction and of the TLR4 function in vitro, we sought a genetic model to definitively address the potential role of Neu1 in the host response to endotoxemia. However, mice with homozygous deletion of the *Neu1* locus develop clinical abnormalities reminiscent of early-onset sialidosis in children, including severe nephropathy, progressive edema, splenomegaly, kyphosis and urinary excretion of sialylated oligosaccharides (*de Geest et al., 2002*). To avoid these developmental defects, we produced bone marrow chimeric mice by reconstituting WT congenic mice with WT or *Neu1$^{-/-}$* bone marrow cells, and compared their survival and cytokine responses following LPS challenge. Bone marrow from CD45.2$^+$ *Neu1$^{-/-}$* donors was as competent as WT bone marrow in reconstituting lethally irradiated CD45.1 hosts (*Figure 4C*), and the major hematopoietic cellular components were not affected by *Neu1* deficiency (*Figure 4D*). Animals reconstituted with *Neu1$^{-/-}$* bone marrow had increased sialylation and Siglec-E binding to DC (*Figure 4E*), and more specifically, Siglec-E-TLR4 association (*Figure 4F*).

To test the role for Neu1 in susceptibility to LPS, we challenged chimeric mice with 100–400 µg LPS/mouse and monitored their survival over a 2 week period. As shown in *Figure 4G*, 100% of mice reconstituted with WT bone marrow cells succumbed to all doses of LPS tested. In contrast, the majority of mice reconstituted with *Neu1$^{-/-}$* bone marrow were resistant to 100 and 200 µg doses of LPS. Corresponding with the increased resistance, the *Neu1$^{-/-}$* >WT chimera mice produced significantly less IL-6 and TNFα following LPs stimulation (*Figure 4H*). These data conclusively demonstrate a critical role for Neu1 in host response to endotoxemia.

## A sialidase inhibitor protects mice against endotoxemia

The significant reduction of endotoxemia in mice with Neu1-deficient hematopoietic cells suggests that endogenous sialidase may be a valuable therapeutic target. To test this concept, we injected 450 µg LPS i.p. into C57BL/6 mice. Immediately after LPS injection, the mice were treated with either PBS or a mixture of two sialidase inhibitors, Neu5Ac2en and Neu5Gc2en, that we previously found to protect mice against polymicrobial sepsis (*Chen et al., 2011*). As shown in *Figure 5A*, while 80% of the vehicle-treated mice succumbed to LPS challenge, all mice that received the inhibitors survived throughout the observation period. Interestingly, the inhibitors had no significant effect on the first cytokine storm, but drastically inhibited IL-6 and TNF-α production 24 and 48 hr post LPS administration (*Figure 5B,C*). Of the two inhibitors used, Neu5Gc2en accounted for most, if not all, of the therapeutic effect (*Figure 5D*).

The fact that sialidase inhibitors have no effect on the first major wave of inflammatory cytokines suggests that the drug can be used therapeutically. Indeed, as shown in *Figure 5E*, Neu5Gc2en conferred complete protection even when it was administered 6 hr after LPS injection. Furthermore, since clinical manifestations are usually caused by a combination of bacteremia and enodoxemia in a more chronic manner (*Hurley, 1995*), we tested the protective effect of NEU1 inhibitor for infection with endotoxin-producing *Escherichia coli*. As shown in *Figure 5F*, Neu5Gc2en protected mice against lethal *E. coli* (strain 25922) infection.

As shown in *Figure 1B,F*, TLR4 interacts strongly with both Siglec-E and F. It is therefore of interest whether the sialidase inhibitor preserved the Tlr4-Siglec-E and/or Tlr4-Siglec-F interactions. We tested whether Neu5Gc2en preserved Tlr4-Siglec interaction using the in vitro capture assay. As shown in *Figure 5G*, LPS treatment reduced both Siglec-E:Tlr4 and Siglec-F:Tlr4 interactions, while treatment with Neu5Gc2en preserved both interactions. Functional redundancy of the two Siglecs explains lack of strong phenotype of mice with Siglec-E deletion in endotoxemia (data not shown).

## Cell surface Neu1 is the therapeutic target of the sialidase inhibitor

As the first step to identify the therapeutic targets of Neu5Gc2en, we compared the inhibitory effect of Neu5Ac2en and Neu5Gc2en against Neu1-4. We transfected 293T cells with cDNA encoding mouse Neu1-4 and use the lysates of the transfectants as the source of sialidases and determined the inhibition of their function by the sialidase inhibitors Neu5Ac2en and Neu5Gc2en. As shown in *Figure 6A*, while both Neu1 and Neu2 were efficiently inhibited by Neu5Ac2en and Neu5Gc2en, Neu3 and Neu4 are largely resistant. Importantly, while Neu2 was equally sensitive to both inhibitors, Neu1 was 10-fold more sensitive to Neu5Gc2en than to Neu5Ac2en. Since Neu5Gc2en but not Neu5Gc2en conferred a

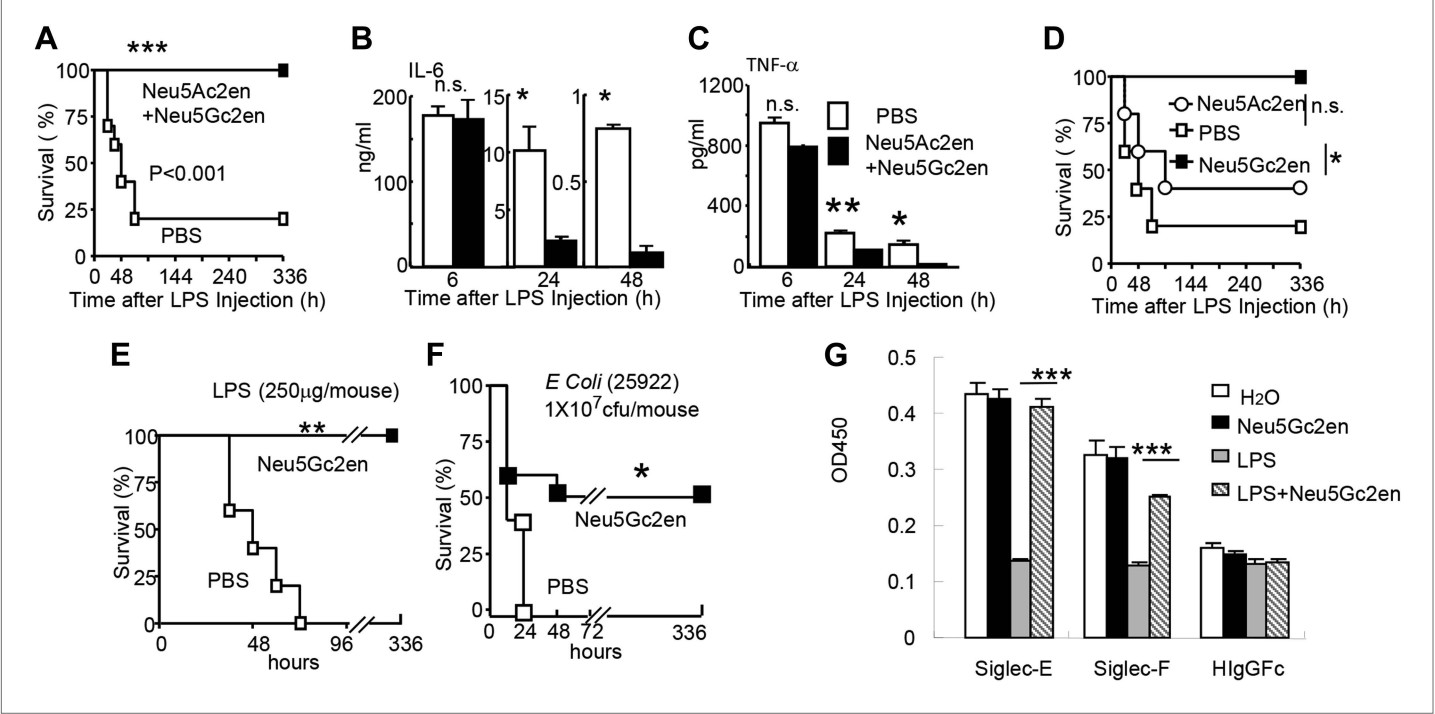

**Figure 5**. Sialidase inhibitors protect mice against endotoxic shock and preserve Siglec-TLR interactions. (**A**) and (**D**) Survival analyses of mice that were treated with 450 μg/mouse of LPS (i.p., *Escherichia coli* 0111:B4). The mice received NeuAc2en and/or NeuGc2en (100 μg/mouse/injection) immediately after LPS administration and every 24 hr thereafter. (n = 8–10 for a and n = 5–6 mice for (**D**), 6–8 week old male mice were used). (**B**) and (**C**) Sialidase inhibitors reduce the levels of IL-6 (**B**) and TNF-α (**C**), measured at indicated time after LPS treatment. Data shown are means ± SD. (n = 8–10, 6–8 week old male mice). (**E**) Therapeutic effect of sialidase inhibitors. C57BL/6 mice received 250 μg/mouse of LPS (i.p., *E. coli* 0111:B4). The mice received NeuGc2en (100 μg/mouse/injection) 6 hr after LPS administration and every 24 hr thereafter. (n = 8–10, 6–8 week old male mice). (**F**) Neu5Gc2en protects mice against lethal *E. coli* (strain 25,922, 10[7] CFU) infection. (**G**) Sialidase inhibitor prevents LPS-induced disruption of the Siglec-TLR4 interaction. 1 × 10[7] splenocytes were cultured in RPMI supplemented with 10% FBS, stimulated with or without 2 μg/ml LPS in the presence of 100 μg/ml NeuGc2en or vehicle for 16 hr, and then the cells were lysed. The lysates were added to wells precoated with Siglece-Fc, Siglecf-Fc or hIgG-fc. The amounts of TLR4 captured were measured using anti-TLR4 mAb. All data in this figure are representative of two to three independent experiments.

therapeutic effect (*Figure 5D*), we hypothesized that Neu1 is likely the relevant target. To test this notion, we compared DC cultured from WT and *Neu1[−/−]* bone marrow for their cytokine response to LPS in the presence or absence of Neu5Gc2en. As shown in *Figure 6B*, targeted mutation of *Neu1* reduced production of both IL-6 and TNFα by more than 70%. Importantly, while WT DC responded to Neu5Gc2en inhibition, *Neu1[−/−]* DC were largely resistant, confirming that Neu5Gc2en inhibits cytokine responses by targeting Neu1.

To ascertain if NEU1 is the therapeutic target of Neu5Gc2en in endotoxic shock, we challenged bone marrow chimeric mice with a lethal dose of LPS (200 μg for WT chimera and 400 μg for *Neu1[−/−]* chimera, see *Figure 4G*) and determined the effect of Neu5Gc2en, administrated at 6 hr after LPS challenge. As shown in *Figure 6C,D*, Neu5Gc2en was therapeutic only in WT, but not *Neu1[−/−]* chimeric mice. Therefore, Neu5Gc2en protects mice against endotoxemia by targeting Neu1.

While Neu1 normally resides in the lysosome, it has been shown to translocate to the plasma membrane after LPS stimulation (*Liang et al., 2006*). To test whether the Neu5Gc2en inhibits cell surface or intracellular Neu1, we first incubated either unstimulated or LPS-stimulated D2SC cells with either Neu5Gc2en or vehicle control for 1 hr and washed away unbound inhibitor. The cells were then lysed to measure sialidase activity. Regardless of LPS stimulation, all sialidase activity was eliminated by shRNA silencing of *Neu1*, which indicated that this assay measured primarily Neu1 activity in the D2SC cells (*Figure 7A*). Upon LPS stimulation, the total Neu1 activity was increased, and all activity was inhibited by Neu5Gc2en (*Figure 7A*). In contrast to LPS-stimulated D2SC, a depot of Neu1 resistant to extracellularly-administered Neu5Gc2en was present in the unstimulated D2SC. However, this residual sialidase activity was eliminated by adding Neu5Gc2en to the lysates (*Figure 7A*). Since unstimulated

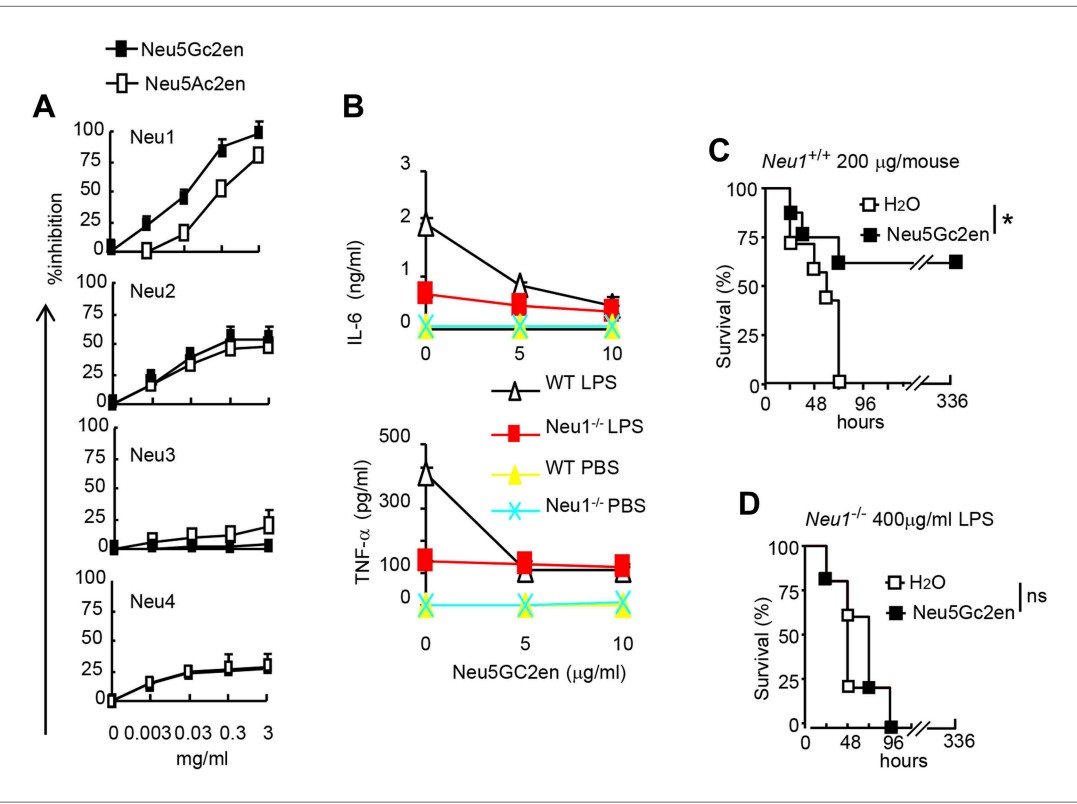

**Figure 6**. NeuGc2en targets Neu1 to inhibit inflammation and confer protection against endotoxemia. (**A**) Comparison of Neu5Ac2en and Neu5Gc2en for inhibitory activity against mouse Neu1-4. Lysates from 293T cells transiently transfected with murine Neu1-4 cDNA were assayed for sialidase activity in the presence of indicated concentrations of inhibitors. Data shown are means ± SD of % inhibition of the activity of each sialidase by indicated concentration of inhibitors. (**B**) Neu5Gc2en targets Neu1 to inhibit production of inflammatory cytokine by DC in response to LPS. TNFα (upper panel) and IL-6 (lower panel) production by DC cultured from *Neu1*[+/+] or *Neu1*[−/−] bone marrow. The DC were stimulated with LPS (100 ng/ml) in the presence or absence of indicated doses of Neu5Gc2en. Data shown are means ± SD (n = 3) and have been reproduced twice. (**C**) and (**D**) Chimeras consisting of either WT or *Neu1*[−/−] bone marrow were challenged with lethal doses of LPS (200 µg/mouse) for WT (**C**), and 400 µg/mouse for mutant chimeras (**D**). Sialidase inhibitor was injected at 6 hr after LPS challenge. Data shown are Kaplan–Meier survival curves. Data are representative of those obtained from two to four independent experiments. Statistical significance of survival analysis was determined using log-rank tests, while the pairwise comparison was performed with Student's *t* tests.

DC also have cell surface Neu1 (*Figure 3C*), we suspect that Neu5Gc2en-sensitive Neu1 resides on the plasma membrane, while the Neu5Gc2en-resistant fraction is intracellular. To test this notion, we fractionated plasma membrane and cytoplasm from D2SC after the live cells were incubated with either vehicle or Neu5Gc2en, and compared their sialidase activities. As shown in *Figure 7B*, while the plasma membrane sialidase activity was significantly reduced, the cytoplasmic sialidase activity was unaffected. Since Neu5Gc2en selectively inhibited cell surface Neu1, it is possible to preserve intracellular Neu1 function while targeting cell surface Neu1 with Neu5Gc2en.

## Discussion

Siglecs are cellular receptors for sialic acid-decorated biomolecular structures. Consistent with its intracellular ITIM- or ITIM-like motifs, engagement of Siglec-G with its natural ligands has been shown to inhibit both innate immune responses and adaptive T cell responses (*Chen et al., 2009*; *Bandala-Sanchez et al., 2013*; *Toubai et al., 2014*).

The function of Siglec 2 (CD22) and Siglec-G in immune tolerance has also been demonstrated using both synthetic ligands and mice with deletions of the Cd22 and *Siglecg* genes (*Duong et al., 2010*; *Jellusova et al., 2010*; *Pfrengle et al., 2013*). Targeted mutation of *Siglecg* revealed its critical

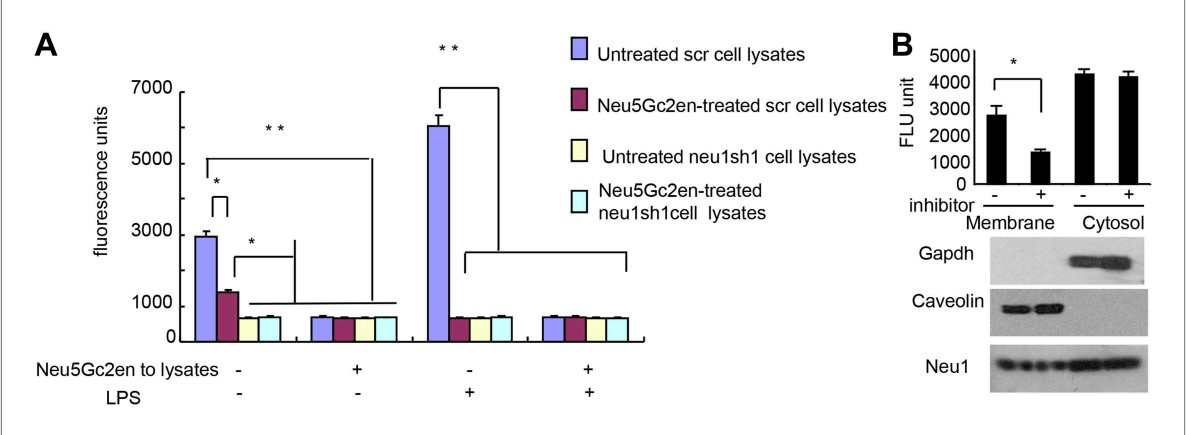

**Figure 7**. Exogenously added Neu5Gc2en inhibits cell surface but not intracellular Neu1. (**A**) LPS stimulation increased sensitivity of Neu1 to exogenously added Neu5Gc2en. D2SC were transduced with lentiviral vectors encoding either scrambled or *Neu1* shRNA and incubated with or without 1 µg/ml Neu5Gc2en for 30 min. After washing away the inhibitors, the lysates were analyzed for residual sialidase activity. Sialidase activity in the lysates was detected with 4-MU-NANA. To confirm that remaining activity is susceptible to Neu5Gc2en, the lysates were also assayed in the presence or absence of 1 µg/ml Neu5Gc2en. (**B**) When added to intact cells, Neu5Gc2en inhibits cell surface but not intracellular Neu1. D2SC cells were incubated with 1 µg/ml of Neu5Gc2en for 1 hr. The plasma membrane and cytoplasmic membranes were prepared as described (***Liu and Fagotto, 2011***) and measured for sialidase activity. The top panel shows Neu1 activity in cell membrane and cytosolic fractions of the D2SC cells (triplicate data, and mean ± SEM), while the lower panel shows commonly used marker proteins to show the purities of the plasma membrane (caveolin) or GAPDH (cytoplasm). Data are representative of those obtained from two independent experiments. Pairwise comparison was performed with Student's *t* tests.

role for inflammatory cytokine production to tissue injuries (***Chen et al., 2009***) and type I interferon response to RNA viruses (***Chen et al., 2013***). Likewise, mutation of *Siglece* has been shown to enhance inflammatory cytokine production to bacterial infection (***Chang et al., 2014***) and neutrophil recruitment to lung (***McMillan et al., 2013***). A fusion protein consisting of extracellular domain of Siglec-E and IgG-Fc (Siglec-E-Fc) is a potent inhibitor of TLR4 response in vitro (***Boyd et al., 2009***). However, the target of Siglec-E-Fc was not identified. Despite these interesting observations, it is unclear how pattern recognition by Siglec family members may relate to other prototypic innate pattern recognition receptors such as TLRs.

By showing a broad physical interaction between Siglecs and TLRs, our data provides a missing link between these two PRR families. The biological significance of the interactions is confirmed by the broad impact of *Siglece* mutation on response of DCs to all TLR ligands tested. Although a number of intracellular negative regulators have been described to regulate TLR function (***Kawai and Akira, 2010***), to our knowledge, Siglecs constitute the first class of cell surface pattern recognition receptors that are directly involved in regulation of TLR function.

By virtue of the broad and direct interaction between TLRs and Siglecs, one may envision that under steady-state conditions, TLR-induced inflammation may be relatively moderate as TLR functions are either directly (in case of Siglec-E) or indirectly (in case of Siglec-G) restrained by sialoside-based pattern recognition. In case of tissue injury, a moderate inflammation may be beneficial for tissue remodeling and regeneration (***Takahashi et al., 2008***). Infection can boost inflammation through microbial sialidase, as we have recently reported (***Chen et al., 2011***), or as shown herein, by inducing translocation of host Neu1 to the cell surface to disarm the Siglec-mediated negative regulation of TLR function.

Since the Neu1 translocation is induced by microbial TLR ligand, a positive feedback is created to release the Siglec brake for TLR activation. The molecular mechanism for TLR4-induced Neu1 translocation is largely unclear, although Neu1 translocation during differentiation from monocytes to macrophage occurs by a route used by MHC class II (***Liang et al., 2006***). Engagement of TLR4 also activates cell surface Neu1 activity, perhaps by G protein-coupled receptor and matrix metalloproteinase-9-dependent mechanisms (***Amith et al., 2009***; ***Abdulkhalek et al., 2011***). In addition to regulating Siglec-TLR interaction, Neu1 has also been shown to induce TLR4 dimerization and thus potentially stimulate TLR4 activity (***Amith et al., 2010***).

It is also of interest to note that TLR-mediated trans-regulation of cellular glycosylation was described through genetic screen in *Drosophila* (*Seppo et al., 2003*), although the functional significance was not clear. The positive feedback described herein suggests a physiological function of TLR-induced glycosylation switch.

Our data showed that Siglec-E mutation affects the function of both cell surface and endosomal TLRs. It is unclear how Siglec-E may affect endosomal TLRs. However, As a member of CD33-like Siglecs that are known to be readily endocytosed (*Walter et al., 2008*), Siglec-E may form physical complexes with endosomal TLRs.

The Siglec family of PRR may help the innate immune system to discriminate between infection and tissue injury by two distinct mechanisms. First, as represented by Siglec-G/10, which does not directly associate with TLR, Siglecs may selective repress inflammation by binding to a DAMPs-associated natural ligand, such as CD24, to inhibit host response to DAMPs. Second, as represented by Siglec-E which directly associates with TLR, Siglecs may regulate inflammation to endogenous TLR ligands by default. By either inducing translocation of intracellular sialidase or by producing microbial sialidases, infection can exacerbate inflammatory cytokine production to disrupt direct Siglec-TLR interactions (*Figure 8*).

The broad impact of Siglecs in the inflammatory response suggests that the interaction with TLR may be preserved to reduce inflammation in cases of detrimental inflammation, such as sepsis. However, since one Siglec can interact with multiple TLRs, and each TLR with multiple Siglecs, it is unlikely that blocking specific Siglecs may significantly impact the susceptibility to endotoxemia. In contrast, as sialidases may regulate multiple Siglec-TLR interactions, targeting host sialidase may be more effective. Indeed, we showed that inhibition of Neu1 by either gene deletion or sialidase inhibitor administration confers a strong protection against endotoxemia. Therefore, Neu1 may serve as a new therapeutic target for endotoxic shock.

Sepsis remains a major challenge despite advances in antibiotics, and many sepsis cases are caused by gram-negative bacteria (*Hurley, 1995*; *Angus et al., 2001*; *Seyrantepe et al., 2003*; *Dombrovskiy et al., 2007*). We have recently shown bacterial sialidase as a valuable target for polybacterial sepsis (*Chen et al., 2011*). However, most pathogens do not encode sialidase. With the identification of a critical function of endogenous sialidase, we can now extend the potential of this strategy to more sepsis-causing pathogens. Unlike bacterial and viral sialidases, endogenous sialidases may be easier to target as they are both limited in diversity and unlikely to acquire drug resistance through mutation. However, since mutation of Neu1 cause sialidosis (*Seyrantepe et al., 2003*), it is of interest to consider how potential side-effects associated with Neu1 inactivation may be avoided. Fortunately, our data demonstrate that an effective sialidase inhibitor can confer protection by selectively targeting cell

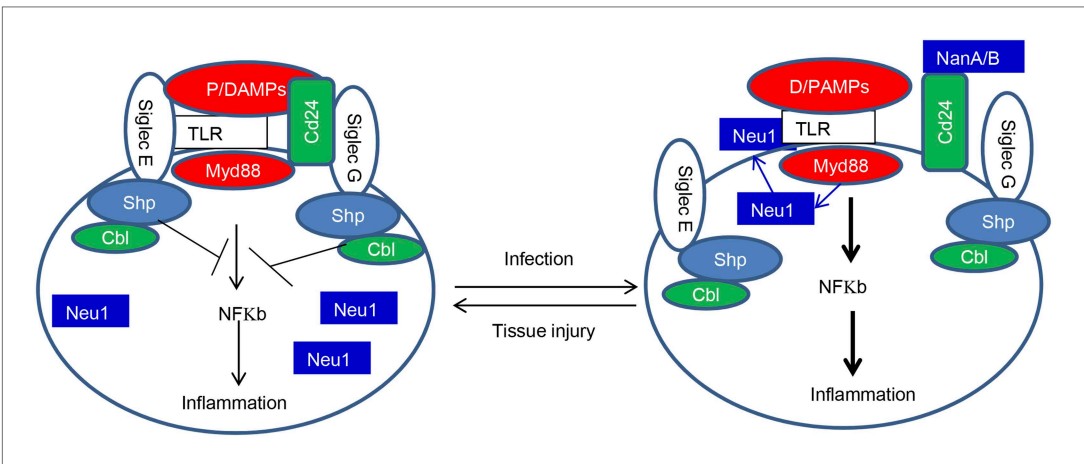

**Figure 8**. Sialoside-based pattern recognition and self-nonself discrimination by the innate immune system. TLR signaling is restrained by Siglecs that are either directly (such as Siglec-E) or indirectly through Cd24 (such Siglec-G) in the case of tissue injuries. Infections cause a positive feedback in TLR signaling. Infections cause translocation of Neu1 to cell surface and/or to production of bacterial/viral sialidases. Both host and cellular to desialylate TLR and/or CD24. The Siglecs are dissociated from TLR to allow a more robust inflammation.

surface Neu1, perhaps by virtue of poor drug accessibility to the intracellular compartment. Therefore, it is feasible to target Neu1 therapeutically without side effects associated with disruption of the intracellular sialidase activity, the known cause of sialidosis.

Taken together, our data establish a missing link between the TLR and Siglec families of pattern recognition receptors. The specificity of individual Siglecs allow them to play distinct roles in innate immunity, while regulation of the interaction by host sialidase suggest a novel approach for treatment of inflammatory diseases, such as sepsis.

# Materials and methods

## Reagents

Recombinant proteins consisting of human IgG Fc and extracellular domains of human SIGLEC1, 2, 3, 5, 6, 7, 9, 10, 11, and mouse Siglec1, 2, E, F were purchased from R&D Systems (Minneapolis, MN). Recombinant mouse Siglec3-hIgGFc was purchased from Sino Biological, Inc. (Beijing, China) and anti-mouse TLR4 (MTS510) from Biolegend (San Diego, CA). Anti-human TLR4 and Anti-Siglec-E were from R&D. Anti-mouse CD11c, CD11b, CD4, CD8, B220 and Gr1 were purchased from eBioscience (San Diego, CA). Anti-Neu1 (H-300) and anti-Neu3 (M-50) antibodies and Horseradish perioxidase conjugated anti-mouse, anti-goat or anti-rabbit secondary-step reagents, as well as anti-TLRs antibodies that are cross-reactive for mouse and human TLR1 (H-90), TLR2 (A-9), TLR3 (M-300), TLR5 (M-300), TLR6 (N-18), TLR7 (N-20), TLR8 (H-114), TLR9 (H-100) and TLR10 (V-20) were purchased from Santa Cruz Biotechnology (Santa Cruz, CA) and biotinylated for studies used in *Figure 1A,B*. Lipopolysaccharide (LPS, from *E. coli* 0111:B4) was from Sigma. *E. coli* (strain 25922) was obtained from ATCC (Manassas, VA). Neu5Ac2en and Neu5Gc2en were synthesized as described (*Li et al., 2008*, *2011*).

## Cell culture and lentiviral infection

D2SC cell line was obtained from Dr Yong-Jun Liu, and maintained in Dulbeco's minimal essential medium supplied with10% fetal calf serum and 1% penicillin and streptomycin. The lentiviral vectors expressing Neu1 shRNAs or Neu3 shRNA were from Thermo Scientific (San Diego, CA). Puromycin was purchased from Sigma. Stable clones were obtained after selection with puromycin (2.5 μg/ml) for 3 weeks after infection.

## Experimental animal models

All mice were used at 6–8 weeks of age. All animal procedures were approved by the Animal Care and Use Committee of University of Michigan and that of Children's National Medical Center.

## *Siglece*⁻ᐟ⁻ mice

Mice with targeted mutations of *Siglece* were derived from 129/Sv ES cells produced by Mutant Mouse Regional Resource Center (MMRRC) at UC Davis (Davis, CA). The mice were backcrossed to C57BL/6 for three generations. Bone marrow from *Siglece*⁺ᐟ⁺ and *Siglece*⁻ᐟ⁻ littermates were used to prepare DC for the current studies. Since the 129/Sv has a mutation in the *Caspase11* gene which is involved in the LPS response (*Broz et al., 2012*), we also typed the genotype of *Caspase11* and excluded mice homozygous *Caspase11* mutation from the current study.

## Bone marrow chimeric mice

Bone marrow chimeras were produced as described (*Chen et al., 2008*), using a total of $5 \times 10^6$ bone marrow cells from either WT or *Neu1*⁻ᐟ⁻ mice as donors, and WT mice as recipients.

## Quantitative real-time PCR analysis

Neu1-4 and TLRs expression measured by real-time polymerase chain reaction. The primers for human TLR were from the TLR-primer set (InvivoGen, version #09H31-MM, Toulouse, France); those for murine Neu1-4 were:

    Neu1 Sense ATGTGACCTTCGACCCTGAG
    Neu1 antiSense TCCTTCTGCCAGGATGTACC
    Neu2 Sense GCTCTACCTGAAGAAGCAGAAG
    Neu2 antiSense GACATGGATTCATGGAGCGGTG
    Neu3 Sense TGCGTGTTCAGTCAAGCC
    Neu3 antiSense GCAGTAGAGCACAGGGTTAC

Neu4 Sense TGGTCTGCGGAGCCTGATATTG
Neu4 antiSense AGTAACGCAGGCACACGGTAG
Hprt Sense AGCCTAAGATGAGCGCAAGT
Hprt antiSense TTACTAGGCAGATGGCCACA

Samples were run in triplicate, and the relative expression was determined by normalizing expression of each target to the endogenous reference, hypoxanthine phosphoribosyltransferase (Hprt) transcripts or GAPDH.

## Construct of plasmids

To generate a construct expressing recombinant proteins consisting of human IgG Fc and extracellular domains of Siglec-G or H, the corresponding cDNA fragment was amplified by PCR and subcloned into expression vector pFUSE-hIgG1-Fc (Invivogen). cDNA for Neu1-4 were amplified by RT-PCR and subcloned into expression vector pCDNA6 (Life technologies, Grand Island, NY). All constructs were verified by restriction enzyme digestion and DNA sequencing. For purification of Siglec-G-Fc or Siglec-H-Fc, the corresponding expression vector was co-transfected with a GFP expression vector into 293 T cells, and stable clones were obtained after 2 weeks culture in selection medium containing 2.5 µg/ml puromycin and 50 µg/ml Zeocin. The stable clones were amplified and cultured in serum free medium, Siglec-G-Fc or Siglec H-Fc was purified with a protein A column from the cell culture supernatants.

## Preparation of DC from murine bone marrow

Bone marrow cells were isolated from femurs and incubated in RPMI complete medium supplemented with 10% fetal calf serum (Life Technologies) and 10 ng/ml recombinant mouse GM-CSF (Peprotech, Coconut Creek, Florida) and 1 ng/ml IL-4 (Peprotech) for 12 days and then stimulated with different TLR ligands from the Mouse TLR1-9 Agonist Kit, InvivoGen. Cytokines in the supernatant were determined using mouse inflammation CBA kit (552364; BD Biosciences, San Diego, CA).

## Flow cytometry

Spleen cells from untreated WT mice, LPS treated WT mice, Neu1 knockout mice or culture cells were washed in flow staining buffer (1× PBS, 2% BSA), and incubated for 1 hr on ice with different directly conjugated-antibodies in flow staining buffer. The intensity of cell-bound antibodies was analyzed on a FACSCanto II cytometer (Becton Dickinson, San Diego, CA). For Siglec-E binding assay, 1 µg of SiglecE-Fc incubated with the cells in 100 µl of the flow staining buffer. Unbound fusion proteins were washed and then incubated with PE-anti-mouse IgGFc antibody (1:400) for another hour on ice. The amounts of cell-bound Siglec-E-Fc was analyzed in Canto II cytometer.

## Immunoprecipitation and immunoblotting

D2SC cell lysates were prepared in lysis buffer (20 mM Tris–HCl, 150 mM NaCl, 1% Triton X-100, pH 7.6, including protease inhibitors, 1 µg/ml leupeptin, 1 µg/ml aprotinin and 1 mM phenylmethylsulfonyl fluoride), sonicated, centrifuged at 13,000 rpm for 5 min and then diluted in IP buffer (20 mM Tris–HCl, 150 mM NaCl, pH 7.6, including the protease inhibitors as described above). Samples were pre-cleared with 60 µl of protein A-conjugated agarose beads (Upstate, Lake Placid, NY) for 2 hr at 4°C or 37°C, and then incubated with corresponding antibodies. Immunoprecipitates were washed four times with IP buffer and re-suspended in SDS sample buffer for Western blot analysis. Coimmunoprecipation between Neu1 and TLR4 were carried out after the live cells were crosslinked. Briefly, D2SC cell lines were stimulated with 2 µg/ml LPS or vehicle for 16 hr. Live cell suspension were incubated with with 1 mM DSP in the reaction buffer (pH7.5, 20 mM HEPES, 0.1 M phosphate, 0.15 M NaCl) at room temperature for 30 min and then stopped by adding 20 mM pH 7.5, Tris and incubated at room temperature for 15 min prior to lysis.

## Siglec-TLR capture assay

96-Well plates were coated with either Siglecs or IgGFc in 50 mM carbonate/bicarbonate buffer, pH 9.5, overnight at 4°C. Wells were blocked with ELISA buffer (20 mM Tris–HCl, 2% bovine serum albumin, 150 mM NaCl, pH 7.6) for 1 hr. Spleen or THP-1 cell lysates were prepared in the lysis buffer (20 mM Tris–HCl, 1% Triton X-100, 150 mM NaCl, PH 7.6), sonicated, centrifuged at 13,000 rpm for 5 min and then diluted in ELISA buffer. 100 µl cell lysates (1 µg/µl) were added to the plate and incubated for 2 hr. Between incubations (all at 37°C), the plates were washed five times with the ELISA buffer. Biotinylated-anti-TLR antibody (0.05 µg/ml) was used to detect bound TLRs. The plate-associated

biotinylated proteins were detected by horse-radish perioxidase (HRP)-conjugated streptavidin (1:1000) for 1 hr and developed with 100 μl/well p-nitrophenyl phosphate liquid substrate system. Absorbance at 450 nm was recorded. In some experiments, cell lysates were replaced with recombinant TLR4 ectodomain (1 μg/ml, R&D) in order to measure direct interaction between Siglecs and TLR4.

## Measurement of inflammatory cytokines

Blood or cell culture supernatants were obtained at indicated times and cytokines in the serum or cell culture supernatants were determined using mouse cytokine bead array designed for inflammatory cytokines (552364; BD Biosciences).

## Neuraminidase activity assay

Sialidase activity was measured using 2′-(4-methylumbelliferyl)-α-D-N-acetylneuraminic acid sodium salt hydrate (4-MU-NANA) (M8639; Sigma) as the substrate. 293 T cells ($4 \times 10^6$) were transiently transfected with 10 μg of Neu1-4 expression vectors as described above or empty vector as control. 48 hr after transfection, cells were harvested and suspended in 100 μl of lysis buffer (20 mM Tris–HCl, 1% Triton X-100, 150 mM NaCl, pH 7.6), sonicated and centrifuged at 13,000 rpm for 5 min. For one reaction, 5 μl of the supernatant was first mixed with different amounts of sialidase inhibitors and then incubated with 4-MU-NANA (final concentration, 15 μM) for 30 min at 37°C in 50 μl reaction buffer (50 mM Sodium phosphate, pH 5.0). The reaction was terminated by adding 600 μl stop buffer (0.25 M glycine-NaOH, pH 10.4) and then fluorescence intensity was measured with a FLUOstar OPTIMA (BMG LABTECH GMBH, Germany) (excitation at 360 nm; emission at 460 nm).

## Immunofluorescence

D2SC cells were seeded in the chamber slides and treated with 100 ng/ml LPS for 18 hr and then fixed with methanol/acetone (50:50 Vol) for 20 min at −20°C. After washing with PBS, cells were blocked by PBS containing 5% bovine serum albumin and stained by rabbit anti-Neu1 polyclonal antibody (H-300; Santa Cruz Biotech) for 1 hr at room temperature. After washing with PBS, cells were stained by Alexa Fluor 568-conjugated anti-rabbit IgG (Invitrogen, San Diego, CA). Subsequently cells were stained by rat anti-TLR4 monoclonal antibody (Sa15-21; BioLegend) and washed and stained by Alexa Fluor 488-conjugated anti-rat IgG (Invitrogen). DNA was stained with DAPI. Fluorescence images were taken under an Olympus X51 microscope.

## Statistical analysis

The differences in cytokine concentrations were analyzed by the Student's $t$ test. The differences in survival rates were analyzed by Kaplan-Meier plot and statistical significance determined using a log-rank test. $p < 0.05$, $p < 0.01$, $p < 0.001$.

## Acknowledgements

This study is supported by grants from National Institutes of Health (AI64350 to YL, AG036690 to PZ, and AI105727 to GYC). Part of the study was performed while some of the authors were at the University of Michigan. The authors have no financial conflict of interest.

## Additional information

### Funding

| Funder | Grant reference number | Author |
|---|---|---|
| National Institute of Allergy and Infectious Diseases | AI64350 | Yang Liu |
| National Institute on Aging | AG036690 | Pan Zheng |
| National Institute of Allergy and Infectious Diseases | AI105727 | Guo-Yun Chen |
| National Institute of Allergy and Infectious Diseases | AI108115 | Xi Chen |

The funders had no role in study design, data collection and interpretation, or the decision to submit the work for publication.

## Author contributions

G-YC, Conception and design, Acquisition of data, Analysis and interpretation of data, Drafting or revising the article; NKB, Acquisition of data, Drafting or revising the article, Contributed unpublished essential data or reagents; WW, Acquisition of data, Contributed unpublished essential data or reagents; ZK, HY, Synthesized sialidase inhibitors Neu5Ac2en and Neu5Gc2en, Contributed unpublished essential data or reagents; XC, Drafting or revising the article, Contributed unpublished essential data or reagents; DV, AD'A, Provided bone marrow cells from Neu1−/− mice and their WT littermates, Advised on established on bone marrow chimera mice, Contributed unpublished essential data or reagents; PZ, YL, Conception and design, Analysis and interpretation of data, Drafting or revising the article

## Ethics

Animal experimentation: This study was performed in strict accordance with the recommendations in the Guide for the Care and Use of Laboratory Animals of the National Institutes of Health. All of the animals were handled according to approved institutional animal care and use committee (IACUC) protocols (312-13-05) of the The Children's National Medical Center. Every effort was made to minimize suffering.

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
