## [Decision Letter]

Thank you for sending your work entitled “Broad and direct cross-talk between TLR and Siglec families of pattern recognition receptors and its regulation by Neu1” for consideration at *eLife.* Your article has been favorably evaluated by Tadatsugu Taniguchi (Senior editor) and 3 reviewers, one of whom is a member of our Board of Reviewing Editors.

The Reviewing editor and the other reviewers discussed their comments before we reached this decision, and the Reviewing editor has assembled the following comments to help you prepare a revised submission.

In this manuscript, the authors extensively examined the interaction between TLR and Siglec family members and convincingly demonstrate the important roles of Siglec-E-TLR4 interaction in host innate response, adding new insight to the interplay of these two PRR families in the innate response. The findings are of great importance, with broad interest in the research of innate immune regulation. The experiments are well designed and the data are well presented. However, further revision seems to be necessary, especially including the confocal microscopy overlay data of these translocated molecules to elucidate their movement and interaction directly, and discussing the working model of Siglec-E appropriately in terms of its broad application.

While the authors' model (Figure 7) does not reveal whether a direct interaction between Siglecs and TLRs is mediated through sialic acid on the TLR itself, the combined data suggest that it may be. The authors should repeat in vitro binding experiments with recombinant TLR4 digested with neuraminidase to determine if this abrogates a Siglec-TLR4 interaction in vitro. Alternatively, cells that were used a source of TLR's could be treated with a commercially available sialyltransferase inhibitor to provide a source of TLRs with minimal sialic acid. Another strategy would be to use a Siglec-Fc (ex. Siglec-E or Siglec-9) bearing an arginine mutant that disrupts sialic acid recognition. It would be useful to know, one way or another, whether the interaction in question is sialic acid dependent.

The authors’ conclusion that their mechanism is broadly applicable to the siglec family is overstated. Data supporting their model was acquired primarily for Siglec-E. As such, it would be more appropriate to focus the title and Abstract on this specific case rather than make a broad claim about the entire family.

Minor comments:

1) It's better to include more details and describe more clearly that why Neu1 or Neu family members are introduced and studied in the effects and mechanisms of Siglec-E-TLR4 interaction and Siglec-E-mediated negative regulation of TLR signaling.

2) As TLR signaling induces Neu1 translocation and then inhibits Siglec-E-TLR4 association, it's better to include the confocal microscopy overlay data of these translocated molecules to elucidate their movement and interaction directly.

3) The translocation of Neu1 from lysosome to membrane is very important for the mechanism of TLR4 signal regulation. The authors should cite, describe, or discuss the underlying mechanism responsible for Neu1 translocation induced by TLR4 activation.

---

## [Author Response]

*[…] The experiments are well designed and the data are well presented. However, further revision seems to be necessary, especially including the confocal microscopy overlay data of these translocated molecules to elucidate their movement and interaction directly, and discussing the working model of Siglec-E appropriately in terms of its broad application*.

We have now provided the confocal image in Figure 3 and revised the title and Abstract.

*While the authors' model (*Figure 7*) does not reveal whether a direct interaction between Siglecs and TLR's is mediated through sialic acid on the TLR itself, the combined data suggest that it may be. The authors should repeat in vitro binding experiments with recombinant TLR4 digested with neuraminidase to determine if this abrogates a Siglec-TLR4 interaction in vitro. Alternatively, cells that were used a source of TLR's could be treated with a commercially available sialyltransferase inhibitor to provide a source of TLRs with minimal sialic acid. Another strategy would be to use a Siglec-Fc (ex. Siglec-E or Siglec-9) bearing an arginine mutant that disrupts sialic acid recognition. It would be useful to know, one way or another, whether the interaction in question is sialic acid dependent*.

The suggested data were provided in Figure 5. We have re-emphasized the point in the Abstract.

The authors’ conclusion that their mechanism is broadly applicable to the siglec family is overstated. Data supporting their model was acquired primarily for Siglec-E. As such, it would be more appropriate to focus the title and Abstract on this specific case rather than make a broad claim about the entire family.

We have taken the reviewers’ advice and moderated the title, Abstract, and Discussion. Our data in Figure 1 clearly showed broad interaction between the two family members. The functional data in Figure 2 show the impact of Siglec E mutation on all TLR ligands. Figure 5 shows association between Siglec E and F are affected by sialidase inhibitors. Therefore, we believe the data are solid in showing the broad cross-reactions, which we believe is a cardinal feature of pattern recognitions and thus would like to emphasize in the title. We believe the modified version is well balanced and supported by data presented.

*Minor comments*:

*1) It's better to include more details and describe more clearly that why Neu1 or Neu family members are introduced and studied in the effects and mechanisms of Siglec-E-TLR4 interaction and Siglec-E-mediated negative regulation of TLR signaling*.

We studied host sialidases because there are no pathogen-encoded sialidases in the endotoxemia model. Our data in Figure 3 showed that of all 4 sialidases known in mammals, only Neu1 and Neu3 are detectable in the D2SC cells. Gene silencing data (Figure 3) and drug inhibition profiles implicated Neu1 (Figure 6). The involvement of Neu1 is validated definitively using bone marrow cells from Neu1-deficient mice.

*2) As TLR signaling induces Neu1 translocation and then inhibits Siglec-E-TLR4 association, it's better to include the confocal microscopy overlay data of these translocated molecules to elucidate their movement and interaction directly*.

These data are included in Figure 3.

*3) The translocation of Neu1 from lysosome to membrane is very important for the mechanism of TLR4 signal regulation. The authors should cite, describe, or discuss the underlying mechanism responsible for Neu1 translocation induced by TLR4 activation*.

We have now expanded the Discussion to address this concern.